# Disseminated MAI in an HIV Patient-An Unusual Presentation

**DOI:** 10.3390/medicines10010010

**Published:** 2023-01-05

**Authors:** Joshni Simon, Joella Lambert, Jose Mosco-Guzman, Kaitlyn Dittmer, Alison Stern-Harbutte, Weston Connelly

**Affiliations:** 1Graduate Medical Education, USF Morsani College of Medicine, HCA Healthcare, Largo Medical Center, 201 14th St SW, Largo, FL 33770, USA; 2Medical Education Department, Dr. Kiran C. Patel College of Osteopathic Medicine, Nova Southeastern University, Tampa Bay Regional Campus, 3400 Gulf to Bay Blvd, Clearwater, FL 33759, USA

**Keywords:** AIDS, psoas abscess, HIV, pleural effusion, splenic lesions, ALP elevation

## Abstract

Patients with Human Immunodeficiency Virus (HIV), and especially Acquired Immunodeficiency Syndrome (AIDS), can present in a multitude of ways with a variety of possible pathologies. This can prove to be a challenge to a clinician. The patient, in this case, was found to have disseminated *Mycobacterium-avium-intracellulare* (MAI), despite compliance with antiretroviral therapy (ART), who presented with right upper quadrant pain, isolated elevated alkaline phosphatase, and sepsis. Imaging revealed multiple splenic lesions, bilateral psoas abscesses, abdominal lymphadenopathy, and a large right pleural effusion with a mediastinal shift to the left. Psoas abscesses were drained and the cultures grew acid-fast bacilli. The patient was treated with azithromycin, ethambutol and rifabutin. Classically, MAI infections of patients compliant with ART therapy present with localized disease. This case offers a different presentation of MAI despite compliance with ART therapy.

## 1. Introduction

Disseminated *Mycobacterium-avium-intracellulare* (MAI) is a disease of immunocompromised and normal hosts. It is acquired from the environment by unknown mechanisms, usually entering the body through the lungs, and then spreads to include the reticuloendothelial systems, bone, and, less commonly, the skin. MAI is one of the two pathogens responsible for nontuberculous mycobacterial infections as part of the *Mycobacterium–avium complex*, (MAC). MAI is an opportunistic infection that is commonly seen in immunocompromised patients with low CD4 counts. MAC in individuals with structural lung diseases typically presents as pulmonary infections [1]; however, in those with Acquired Immunodeficiency Syndrome (AIDS), focal lymphadenitis and disseminated disease are the most common manifestations [2,3]. MAI infections can have localized or disseminated presentations. The classic symptoms of disseminated MAI are fever, night sweats, abdominal pain, diarrhea, anorexia, fatigue, and malaise [4,5,6]. Along with the classic presentation of disseminated MAI, it is common to find infections spreading through the bloodstream and to the lymph nodes and liver. With adherence to Highly Active Antiretroviral Therapy (HAART), there has been a decrease in the MAI epidemic; however, nonadherence to HAART can decrease the CD4 count and hence increase susceptibility to MAI infection [7,8,9]. Through the development of HAART and subsequent adherence to the therapy, it has been shown in Human Immuno-deficiency Virus (HIV) infection-positive individuals to postpone the progression of MAI infection [6]. Immunocompromised patients with localized or disseminated MAC infections are normally treated for at least 12 months with a dual therapy regimen of a macrolide and ethambutol; however, those presenting with constant high numbers of mycobacteria or failing antiretroviral therapy (ART) are given a third agent, such as rifampin, to overcome their infection [2,3,6].

## 2. Detailed Case Description

The patient is a 35-year-old African American male with a past medical history of HIV (diagnosed at 18 years of age), schizophrenia, depression, combative/aggressive behavior, and candida stomatitis who presented to a psychiatric facility for depressive thoughts and suicidal ideation. He was medically cleared for inpatient psychiatry but found to have facial lesions that were thought to be contagious and possibly resemble Herpes Simplex Virus (HSV) or Varicella Zoster Virus (VZV). He was transferred to a medical-behavioral unit due to the need for isolation precautions and the requirement for closer supervision. The original workup was significant for elevated alkaline phosphatase with mild transaminitis. He was also found to have right upper quadrant abdominal tenderness. The lesions identified on his face resembled acne vulgaris with no evidence of other infectious or malignant cutaneous lesions on a full-body examination.

Triple-phase computerized tomography (CT) was obtained for further evaluation of his abdominal pain. Imaging was significant for bilateral psoas abscesses [Figure 1], [Figure 2], multifocal lesions in the spleen [Figure 3], and a large right-sided pleural effusion with mediastinal deviation leftward [Figure 4]. The patient became more septic-appearing with leukopenia, fever, and chills. Due to his change in clinical presentation, blood cultures were drawn, he was started on broad-spectrum antibiotics, and transferred to the main hospital campus with an infectious disease consult. CD4 was 45 despite reported adherence to his HIV medication.

At the main hospital, the patient underwent several procedures for further workup of his condition. This included a biopsy and culture of his psoas abscesses and a thoracentesis with pigtail placement. Cultures returned positive for acid-fast bacilli (AFB) from his psoas abscesses but were negative for *Mycobacterium tuberculosis*. Cultures were negative from his thoracentesis and his pleural effusion, which was exudative with adenosine deaminase at a level of 45. He was treated empirically for disseminated MAI and continued on his home HIV medication. Discussion regarding goals of care with the patient and his family revealed a desire to transfer to a hospice level of care and move closer to his family.

### 2.1. Social History

The patient denied current alcohol use. He performed illicit intravenous drug use in the past. He inhales tetrahydrocannabinol daily, was a former tobacco smoker (5 pack-years), and is a current long-term nursing home resident with assistance in administering medications.

### 2.2. Medications

The patient takes the following medications by mouth: valproic acid 250 mg twice a day, haloperidol 5 mg twice a day, levetiracetam 250 mg twice a day, levothyroxine 50 mcg daily, lorazepam 1 mg twice a day, midodrine 5 mg three times a day, mirtazapine 15 mg at bedtime, morphine 20 mg every 4 h as needed for pain, trazodone 100 mg at bedtime, abacavir-dolutegravir-lamivudine 699 mg-50 mg-300 mg one tab daily.

### 2.3. Vital Signs

Vitals were significant for sinus tachycardia without evidence of fever.

### 2.4. Physical Exam

General: Alert, awake, cachectic, and thin African American male in no apparent distress with severe muscle wasting and minimal subcutaneous fat present

HEENT: Atraumatic, normocephalic, no thyromegaly, no lymphadenopathy

Neck: No masses or swelling, moist mucous membranes

Respiratory: Symmetric expansion, no respiratory distress, no conversational dyspnea, on room air, decreased breath sounds over the right lung field

Cardiovascular: Regular rate and rhythm, normal capillary refill

Abdominal: Rectus abdominis with the minimal overlying fat present, tender in the right upper quadrant with voluntary guarding, no involuntary guarding or rebound tenderness, normal bowel sounds

Extremities: No cyanosis, no pitting edema

Neurology: Alert, normal speech, oriented to person only, in a wheelchair due to gait disturbance, his thin legs were able to move against gravity with 4/5 strength bilaterally, he had full upper extremity strength 5/5 bilaterally

Skin: dry, normal temperature, normal skin turgor

Right chin pustule resembling acne vulgarisDried lips and mouth, chapped skin around the edges of his mouth with his tongue bright red in coloration, no evidence of blisters or lesionsNo other visible lesions on his body including no evidence of rashes

Psychiatry: Abnormal insight and judgment into his medical condition, poor hygiene, psychomotor retardation when responding to questions and performing tasks, limited concentration and attention; no active suicidal ideation, restless.

## 3. Discussion

Among patients with HIV, MAC infection most commonly manifests in those with a CD4 count <50 cells/microliter [10]. There has been a significant decrease in the incidence of MAC due to prophylactic treatment and the effectiveness of antiretroviral therapy [1,8]. In our case, it is valid to question the adherence to antiretroviral therapy due to extensive psychiatric history and features present on the physical exam such as psychomotor retardation. The patient also exhibited frank muscle wasting which initially prompted the concern for a workup of AIDS. The patient was wheelchair-bound due to significant issues with walking which were later found to be attributed to the bilateral psoas muscle infiltration. The patient was able to give an accurate history of both how long he was diagnosed with HIV in addition to an accurate list of his medication, doses, and schedule for taking his medication which supports an accurate insight into his medical condition. Further investigation with the skilled nursing facility revealed that the patient had been adherent to his HIV medication for the last two years since he was residing in the facility (presumably since he had difficulty walking). Prior to his initial hospitalization for lower extremity weakness, it is unknown if the patient was adherent to his antiretroviral treatment. It was postulated but never confirmed through virology that the patient most likely had resistance to the medication he was taking rather than poor adherence, especially since he was not active with an HIV specialist outpatient. It was also likely that the patient had an indolent mycobacterial infection that was never found due to the patient’s non-specific findings of lower extremity muscle weakness and psychiatric history. Unfortunately, disseminated MAI presents with non-specific clinical symptoms of intermittent or persistent fever, night sweats, abdominal pain, weight loss, fatigue, malaise, and anorexia [4,5]. Disseminated MAI would have been difficult to work up and diagnose unless there was a strong suspicion of AIDS (as there was during the initial presentation at our hospital). Ultimately, rifabutin was added to his treatment regimen due to suspected resistance.

Our case report does have limitations in the formal diagnosis as the only isolate found during a thorough workup was acid-fast bacillus that was non-TB in the psoas abscess. Blood, sputum, and pleural cultures were negative for any other organism. The diagnosis of MAC is typically achieved by isolating the organism in culture, usually in blood or lymph nodes [11,12]. His compilation of symptoms was the most consistent with disseminated MAI and it was unlikely given his low CD4 count that it was a contaminant. In addition, there are limited studies on MAC disease in patients that are already well-established and adherent to their antiretroviral agents. Our case report points to the MAC disease progression in a patient that is adherent to ART.

Other cases reveal that it is rare for patients adherent to antiretroviral therapy to have disseminated disease and that it is much more common for them to have localized disease [13,14]. It is unclear if the difference in presentation for MAC infection relies on treatment for HIV (whether it is effective or not based on resistance) or if it is more associated with CD4 level. Patients known to be treatment-naive who initially present with AIDS and MAC infection can have an excellent recovery in CD4 count leading to the resolution of MAC infection.

Another unusual component of this patient’s presentation was the involvement of his gastrointestinal tract. His alkaline phosphatase was elevated to 1427 (Units/L) with AST and ALT elevated at 137 and 77 (Units/L) respectively. At first, there was a concern for drug-induced liver injury as some antiretroviral therapies are cleared through the liver. This prompted imaging of his abdomen and the only modality available at the psychiatric hospital was computed tomography. Based on imaging, it is likely that liver injury, if any, occurred secondary to splenic injury. Disseminated MAI left untreated can result in splenomegaly, multifocal infarction of the splenic parenchyma, and near-total replacement of the splenic parenchyma [15]. The patient most likely suffered this injury to his spleen due to the length of his infection being untreated for over two years and the inability of his immune system to mount a response. His alkaline phosphatase decreased in value dramatically during admission suggesting that there was a strong element of reactivity associated with the laboratory value.

The cornerstone of treatment for MAI is a dual therapy regimen consisting of ethambutol and macrolides [2,3,6]. While it will never be confirmed whether the patient had true antiretroviral resistance as the testing was not available in our hospital and was typically done in an outpatient setting, the patient elected to pursue hospice in lieu of further testing and antimicrobial treatment.

## 4. Conclusions

Disseminated MAI is a very rare disease that can present in a variety of ways and should be considered for any patient with AIDS with a CD4 count of less than 50 [16]. If CD4 count is initially unavailable and adherence to antiretroviral medication is in question or the patient exhibits multiple signs of muscular wasting, effort should be made to complete a full history and physical and direct advanced testing modalities toward any positive findings. More research is needed to capture the extent of the disease and delineate its progression, whether localized or disseminated, especially in how it relates to antiretroviral therapy adherence and resistance.

## Figures and Tables

**Figure 1 medicines-10-00010-f001:**
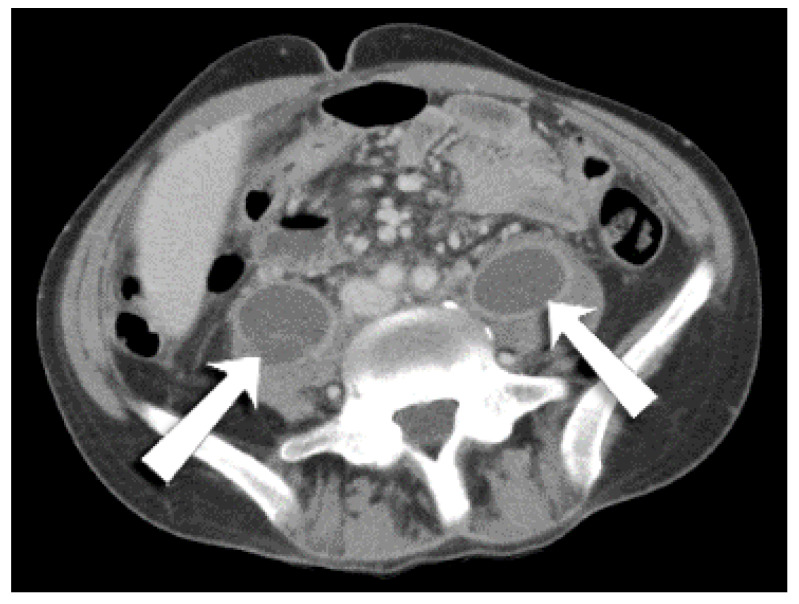
This is an axial view demonstrating bilateral psoas abscesses with ring-enhancement.

**Figure 2 medicines-10-00010-f002:**
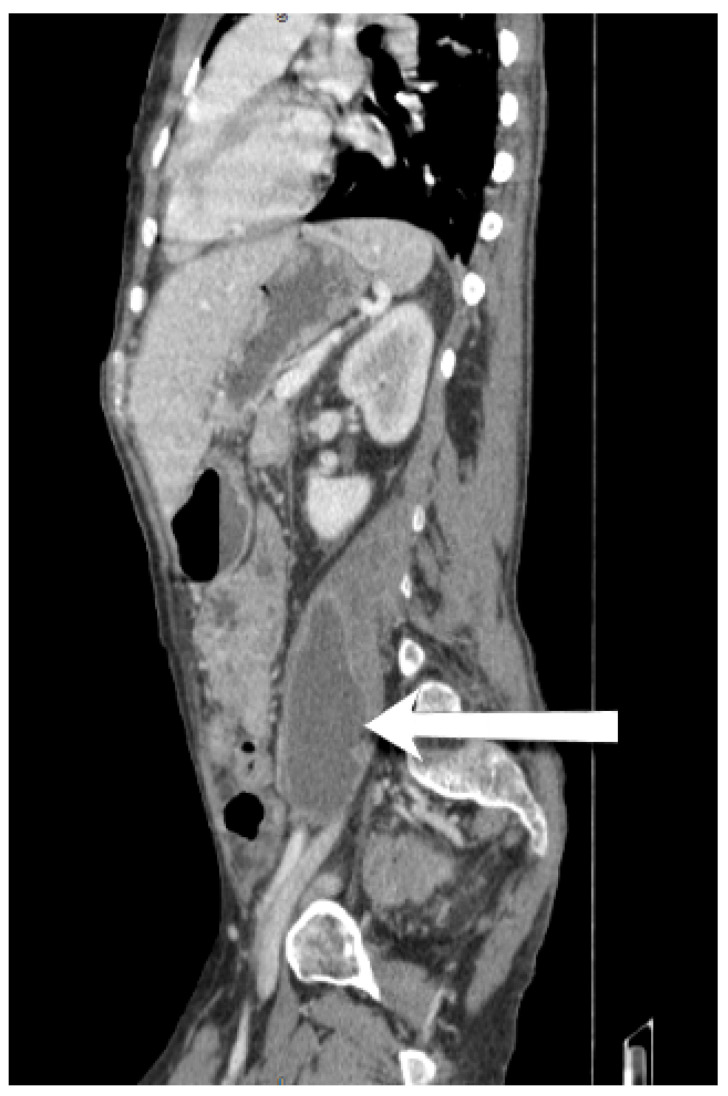
This is a sagittal image demonstrating the length and extent of infiltration of the psoas abscess.

**Figure 3 medicines-10-00010-f003:**
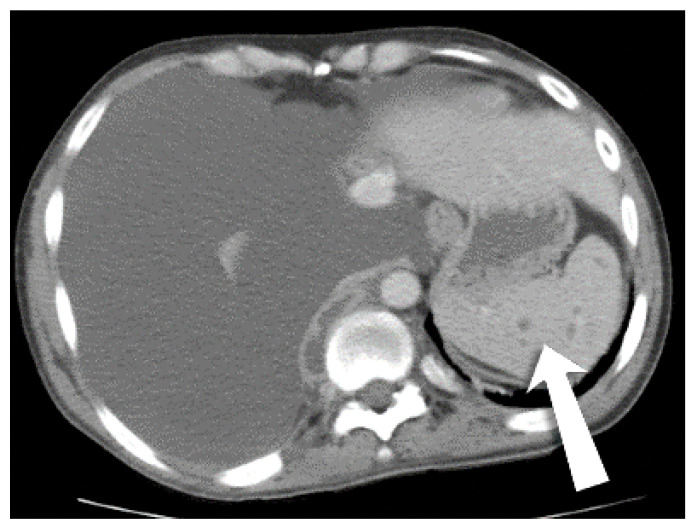
This is an axial view demonstrating the large right pleural effusion as well as multifocal splenic lesions.

**Figure 4 medicines-10-00010-f004:**
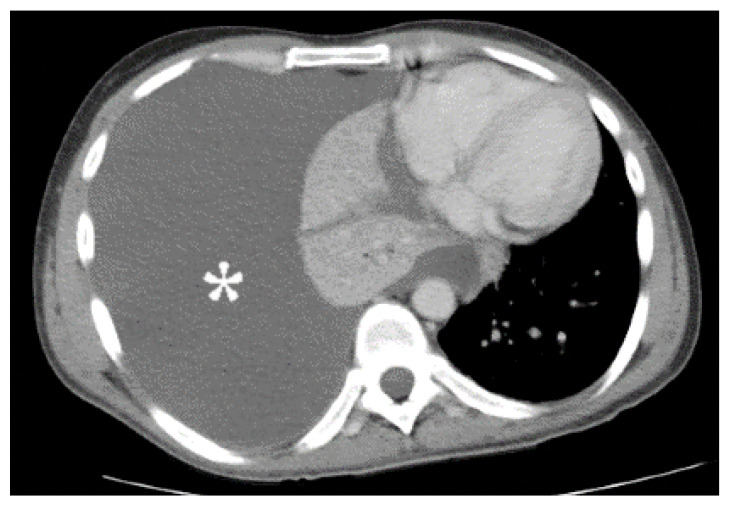
This is an axial view demonstrating a large right pleural effusion with mediastinal shift leftward demarcated by an asterisk.

## Data Availability

Not applicable.

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
