# Peer review of "Disseminated MAI in an HIV Patient-An Unusual Presentation"

_medicines, 2023, doi:10.3390/medicines10010010_

Round 1

Reviewer 1 Report

In this manuscript “Disseminated MAI in a HIV Patient-An Unusual Presentation” the authors present a case study of HIV patient with Disseminated MAI.  MAC is caused by 2 species M. Avium and M. intracellulare and commonly found in immunocompromised patients.  Why does the author think that this patient exhibits unusual symptoms. Was the Mycobacterium-avium-intracellulare (MAI) species confirmed by any molecular biology tool? The Discussion should be improved and not just repetition of case description.

Author Response

Point 1: In this manuscript “Disseminated MAI in a HIV Patient-An Unusual Presentation” the authors present a case study of HIV patient with Disseminated MAI.  MAC is caused by 2 species M. Avium and M. intracellulare and commonly found in immunocompromised patients.  Why does the author think that this patient exhibits unusual symptoms.

Response 1: The patient was more of an unusual presentation rather than presenting as unusual set of symptoms.

Point 2: Was the Mycobacterium-avium-intracellulare (MAI) species confirmed by any molecular biology tool?

Response 2: The only molecular biologic testing that came back positive was acid-fast bacillus. We were never able to isolate MAI species. The only distinction the lab was able to make was non-tuberculosis. We were told that the testing would take 4-6 weeks to come back. Due to this delay, based on the patient's presentation, we believe that the patient had disseminated MAI and he was treated as such. The culture never resulted past acid-fast bacillus. Unfortunately the microbiology lab no longer has the patient's sample. This is a valid point and we did revise the discussion to include the limitation of our testing.

Point 3: The Discussion should be improved and not just repetition of case description.

Response 3: The discussion section was heavily revised.

Reviewer 2 Report

The article is clear in its content and has an adequate sequence.

-However I feel that a clear determination of the presence of MAC in blood or other body fluids is needed, as a basis for diagnosis. What was done by images as well as clinical signs and symptoms should be a complement to the diagnosis.

-All figures need to include arrows or another sign in order to localize easier the mentioned lesion or finding. 

-The Discussion section could be improved in order to compare findings un other study cases and differentiate from the introduction section because similar information is highlighted.  

-To the extent possible use more recent references

Author Response

Point 1: However I feel that a clear determination of the presence of MAC in blood or other body fluids is needed, as a basis for diagnosis. What was done by images as well as clinical signs and symptoms should be a complement to the diagnosis.

Response 1: We were only able to isolate acid-fast bacillus that was non-TB in the psoas abscess. All other cultures were negative. Unfortunately, the sample was never processed further and we were unable to isolate the particular MAI species. The patient's clinical condition and imaging did support a diagnosis of disseminated MAI and the patient certainly had improvement after initiation of therapy. The isolate was never identified 4-6 weeks after collected and due to it being isolated from a fluid sample, I suspect the sample had exceeded its shelf-life prior to being able to identify the isolate. This is certainly a limitation that was added to the discussion section in the most recent revision.

Point 2: All figures need to include arrows or another sign in order to localize easier the mentioned lesion or finding. 

Response 2: The figures were edited to include demarcation in order to localize the lesion in an easier fashion.

Point 3: The Discussion section could be improved in order to compare findings un other study cases and differentiate from the introduction section because similar information is highlighted.  

Response 3: We heavily revised the discussion section in the revision attached.

Point 4: To the extent possible use more recent references

Response 4: We performed a thorough literature search regarding similar case studies and references. Unfortunately (and fortunately) the incidence of disseminated MAI in HIV patients has decreased tremendously due to ART, MAI ppx, and HIV resistance. We were unable to find similar cases for comparison after the early 2000s.

Reviewer 3 Report

Mycobacterium avium intracellulare (MAI) infection is common in Human Immunodeficiency Virus (HIV) patients. MAI infection can show a localized or disseminated presentation. MAI infections of patients compliant with ART therapy present with localized disease. This case report reports a different presentation of MAI infection in an HIV patient despite adherence to ART. 

I think the author did a great job in this case report. It is well written with enough details.

The case report in principle can be accepted after minor revisions or clarifications: 

1-   Confirming that the patient complied with ART therapy

The entire case report idea is based on the fact that the patient complied with ART therapy. The author writes in the social history in lines 65 and 67 that the patient denied current alcohol use and that he is a current long-term nursing home resident with assistance in administering

medications.

It will strengthen the report if they specify some of the following points if possible:

1- When the current long-term nursing home residency started/Specify that he was receiving assistance in getting his ART regularly for x number of years or so.

2- Since when he was adherent to ART

Not sure to what extent we can trust someone’s saying he complied with ART when he presented to a psychiatric facility for depressive thoughts and suicidal ideation and has a history of schizophrenia and combative/aggressive behavior. The patient also showed poor hygiene, psychomotor retardation when responding to questions and performing tasks, and limited concentration and attention. 

Or at least how can we trust that he was ~100% committed to his ART on a regular base? 

Are there specific criteria that assure us that he was entirely in compliance with ART? If so, I think clarifying this will strengthen the report.

2- Degree of transaminitis.

  • Line number 43, and 44, the author wrote: Original workup was significant for elevated alkaline phosphatase with mild transaminitis. 
  • Whereases on lines number 111 and 112, the author wrote: Throughout the course, the patient continued to worsen and reported right upper quadrant abdominal tenderness. Labs were significant for transaminitis.

Maybe the author can say: Lab became significant for transaminitis. To make it flow with the story. Just a suggestion.

 Thank you.

Author Response

Point 1: Confirming that the patient complied with ART therapy

Response 1: You bring up a valid point regarding compliance/adherence. We were able to confirm adherence for the last two years by speaking personally with his care home (SNF) where he came from prior to our hospital. I do believe it is in question that prior to then (two years from when he was first living in a care-home that primarily assisted with ADL/IADLs and administering his medication), he could have had non-compliance and if he was never following with an HIV specialist outpatient, there is a chance the patient was continuing ART that his particular HIV strain was resistant to thus causing MAI infection and such low CD4. Although I would never be able to confirm this, I wonder if he had untreated MAI for several years without workup or follow up. He also could have had treatment for MAI in the past which could have left us with relatively sterile blood and fluid cultures. Maybe he completed treatment for MAI (localized) but due to ART resistance he progressed to disseminated MAI. The patient did come in unable to walk secondary to bilateral psoas abscesses which I doubt happened overnight. Definitely a valid point which we addressed to our discussion section in the newest revision.

Point 2: Degree of transaminitis.

Response 2: I did try to correct this and specifically state the numerical values of ALP/AST/ALT in order to clarify this point in the most recent revision.

Round 2

Reviewer 1 Report

It would have been better if changes were highlighted in the revised version. 

Accept the revised version. 

Reviewer 2 Report

The manuscript was improved according to previous comments, especially the discussion was improved, interestingly, the weaknesses of the work are also discussed.